# The Influencing Factors of Aggregation Behavior of Tree-of-Heaven Trunk Weevil, *Eucryptorrhynchus brandti* (Harold) (Coleoptera: Curculionidae)

**DOI:** 10.3390/insects14030253

**Published:** 2023-03-03

**Authors:** Xuewen Sun, Wei Song, Wenjuan Guo, Shujie Wang, Junbao Wen

**Affiliations:** Beijing Key Laboratory for Forest Pests Control, College of Forestry, Beijing Forestry University, No. 35, Tsinghua East Road, Beijing 100083, China

**Keywords:** coleoptera, pest weevil, aggregation, conspecific communication, environment factors

## Abstract

**Simple Summary:**

The tree-of-heaven trunk weevil *Eucryptorrhynchus brandti* is a significant pest of the tree of heaven *Ailanthus altissima,* often leading to *Ailanthus altissima* death. *Eucryptorrhynchus brandti* adults usually aggregate to overwinter in soil or areas dense with weeds, and congregate on the bare phloem of the tree of heaven in the field during the non-winter months regardless of whether the trees are healthy or injured. Aggregation is a common phenomenon, and conspecific individuals gather in a limited area of the environment and increase social interaction. In this study, some factors were tested in the laboratory, indicating the environmental factors, potential chemical cues, and some physical signals that had the most significant effects on the aggregation behavior of *E. brandti* adults. This study of the aggregation behavior of *E. brandti* adults contributes to understanding communications and interactions among conspecific individuals.

**Abstract:**

The tree-of-heaven trunk weevil, *Eucryptorrhynchus brandti* (Harold) (Coleoptera: Curculionidae), is one of the most harmful pests that damage the tree of heaven, *Ailanthus altissima* Swingle (Sapindales: Simaroubaceae). Aggregation behavior tests of *E. brandti* adults were conducted in laboratory conditions. The effects of temperature and light on the aggregation behavior of adults were tested, and the effect of sex and host was conducted with binomial choice experiments. The results showed that (1) the adults aggregate in both light and dark environments but preferred the dark environment, (2) temperature can drive the aggregation of *E. brandti* adults, (3) host plants could trigger *E. brandti* adults’ aggregation behavior, which is probably related to phytochemicals and insect feeding and localization, (4) there was mutual attraction of males and females and chemical attraction of crude intestinal extracts of males and females, and (5) aggregation behavior of *E. brandti* adults may also be related to the mediating of physical signals in insects. In this study, aggregation behavior can help us understand conspecific interactions and discover some strategies for effective control.

## 1. Introduction

The tree-of-heaven trunk weevil *Eucryptorrhynchus brandti* (Harold) (Coleoptera: Curculionidae) has one generation every year with overlapped generations and weak flight ability [1]. The species is one of the major destructive insect pests of the tree of heaven *Ailanthus altissima* Swingle (Sapindales: Simaroubaceae), a tree with a high ornamental value that plays an important role in China’s urban greening and shelterbelt construction [2,3,4]. *Eucryptorrhynchus brandti* adults feed on the trunk of the tree of heaven, and larvae develop under the bark, destroying the phloem and xylem. *Eucryptorrhynchus brandti* is widely distributed in Beijing, Shandong, Hebei, Ningxia, Liaoning, Jilin, and other regions in China. The harm and the mortality rate of *A. altissima* damaged by *E.brandti* and close relative the weevil *E.scrobicuatus* are steadily rising in the whole second-generation farmland forest network in the Ningxia Hui Autonomous Region, China. Physical controls have been widely used to control *E.brandti* [5,6]. However, most current methods are not effective enough, and developing effective strategies to control this weevil is more urgent. *E. brandti* adults usually aggregate to overwinter in soil or in areas dense with weeds [7], and congregate on the bare phloem of the tree of heaven in the field during the non-winter months, regardless of whether the trees are healthy or injured. Based on these observations, we suggested that some factor objects may be attractive or trigger the aggregation tendency and behavior of the weevil.

Aggregation behavior has been observed in lots of insects such as cockroaches and locusts [8,9,10], and many animals such as Talamanca striped rocket frogs *Allobates talamancae* Cope and Eurasian scops owls *Otus scops* Linnaeus [11,12]. Aggregation is defined as a higher density of individuals than in the surrounding environment, regarded as a strategy to avoid adverse environments [13]. Aggregation behavior is widespread in social insects such as *Lasius niger* Linnaeus and *Schistocerca gregaria* Forskal [14,15], and non-social insects such as *Capnodis tenebrionis* Linnaeus and *Porcellio scaber* Latreille [16,17], and may appear in response to environmental heterogeneities or the attraction between individuals [18].

Heterogeneities in the environment can drive animal aggregation. Animals always choose favorable environments and aggregate in limited space. For example, convergent lady beetle *Hippodamia convergens* Guerin-Meneville individuals prefer suitable temperature zones, and form clusters through the process of individual movement [19]. Some isopods were found aggregated in damp places during dry periods, or at least in places where they were protected from the evaporating power of the air [13]. The presence of environmental heterogeneities reflects individual preferences for the environment, such as a patch of food, light, humidity, or physical constraints of the environment [20,21,22], and these are also affected by resting time, phototaxis, and chemotaxis [23].

The interaction and communication between conspecific animal species can also contribute to the occurrence of aggregation behavior [24,25]. Individuals follow a dynamic signal others provide to produce aggregation [26,27]. These signals comprise particular combinations of chemicals and specific components. Due to the particularity of the chemical composition and specificity, the conspecific insects use chemical signals as a means of communication between individuals [28,29].

Few studies related to the individual interaction of *E. brandti* adult individuals and their response to environmental heterogeneity have been carried out. Here, we conducted choice and non-choice tests to compare the aggregation behavior or aggregation tendency of *E. brandti* reacting to multiple factors: (i) individual sex, (ii) light intensity of the habitat, (iii) the host plant *A. altissima*, and (iv) temperature. This study aims to investigate the effect of environmental factors, individual signals, and the host plant on the aggregation preferences and tendencies of adults of *E. brandti,* and provide a theoretical basis for new ways to control and manage the trunk pest. 

## 2. Materials and Methods

### 2.1. Test Insects

Approved by the Lingwu Natural Resources Bureau, several trees were felled in Lingwu Farm (38°08′14.76″ N, 106°16′52.27″ E), Wudongshu Township (38°11′27″ N, 106°15′59″ E), Wujiahu Village (38°0′7.51″ N, 106°17′54.59″ E), Yinchuan City, Ningxia Hui Autonomous Region, China, and the cut wooden trunks were brought back to the laboratory (T = 22 ± 2 °C, RH = 65 ± 5%, L:D = 14:10 h). *Eucryptorrhynchus brandti* adults were collected when they crawled out from the stems and fed on young branches of *A. altissima* for 3–5 days to adapt to the laboratory environment conditions. 

### 2.2. Binomial Choice Experiment of Host Plant

The experiment was carried out in a glass arena (44 cm × 44 cm × 3 cm) containing two identical shelters placed symmetrically on either side of the center (Figure 1A). A 50 cm × 50 cm square of filter paper was laid under the experimental device, and the filter paper was replaced every time the experiment was conducted. A luminosity of 1600 ± 200 lux was provided by two neon lights of 40 W, which were centered on the experimental arena. Two upside-down glass Petri dishes (diameter: 9 cm; height: 2.5 cm) were used as shelters (Figure 1B). Four holes 1.5 cm in diameter were pierced equidistantly around each dish, allowing *E. brandti* adults to enter the shelters. *Eucryptorrhynchus brandti* adults were released at point C (Figure 1A). One of the two shelters contained fragmented young branches of *A. altissima,* and another was empty as a control. A total of 1800 male and 1800 female adults were tested. Twenty female or male adults were released at a time, and we recorded the number of individuals located in each shelter at 10, 20, 30, 40, 50, 60, 90, 180, and 360 min, repeated 10 times, and replaced the test adults after each test. At the end of each operation, the experimental equipment was cleaned with 75% alcohol in a timely manner. The experiments were performed under the following conditions: 14/10 h light photoperiod, 22 ± 2 °C, and 65 ± 5% RH. 

### 2.3. Temperature Experiment

We explored the aggregation behavior and tendency of *E. brandti* adults under three temperature gradients, low temperature (5–15 °C), ordinary temperature (20–30 °C), and high temperature (30–40 °C), which were adjusted using an electronic semiconductor refrigeration and heating instrument (99 Electronics Company, Guangdong, China). The filter paper was replaced in the glass arena under chamber A and chamber B. A semicircular hole with a radius of 1 cm was cut on each side of chambers A and B, to connect corridor C. A total of 600 males and 600 females were used in the experiment, of which 300 males and 300 females were used to measure aggregation behavior and the remaining 300 males and females were used to measure aggregation tendency. Aggregation degree (the number of clusters formed by two or more individuals) and max aggregation (the number of individuals in the largest cluster) [30] were used to measure the aggregation behavior of adults at different temperatures. Ten males or ten females were placed in three temperature zones: low, ordinary, or high temperature. Aggregation degrees and the max aggregation of *E. brandti* adults at 10 and 90 min were counted and recorded. The experiment was repeated 10 times. Furthermore, the trend of aggregation was detected by observing the number of adults in the low-, ordinary-, or high-temperature zones arriving in the zone of ordinary temperature. Chamber A was kept at a low or high temperature or ordinary temperature, and chamber B was always at an ordinary temperature (Figure 2). Ten males or ten females were placed in chamber A when temperature readings stabilized, and the number of adults entering chamber B from chamber A through the corridor C was recorded at 10 and 90 min. The experiment was also repeated ten times. After each operation, chambers A and B and corridor C were cleaned with 75% ethanol.

### 2.4. Conspecific Individual Interaction

#### 2.4.1. Gender Factor

Thirty females or males as the odor source were placed in the sample bottle connected to the side of the Y-tube olfactory instrument, and the control tube was empty. The odor source was replaced every 2 h to ensure effectiveness and high activity. All steps were operated in a dark environment. The 120 males and 120 females were divided into 6 groups with 20 males and 20 females in each group. Twenty males or females in each group as test subjects were starved for 24 h to avoid interference with residual food. We turned on the atmosphere sampler (QC-IS, Beijing Labor Protection Research Institute, Beijing, China) for 30 s (flow rate 1.5 L/min) and observed the adults for 10 min. When the tested adults reached 1/2 of one of the arms or passed through it completely and stayed on the arm for more than 5 min, it was recorded that the tested adults could make a choice. If the tested adults did not decide within 10 min, the data of these adults were invalid. The inner arm was cleaned with 75 % ethanol and air-dried at ordinary temperature. After testing half of the adults in each group, the positions of the arms on both sides of the Y-tube olfactory instrument were reversed, and the remaining adults were tested by the same method.

#### 2.4.2. Reaction to Crude Extracts

The intestines of 30 males or females treated with starvation were dissected, then soaked in 2 mL n-hexane for 45 min, and concentrated to 500 μL with a nitrogen concentrator to obtain the crude intestinal extract. A small blob of cotton with 50 μL crude extract was placed in the sample bottle as the odor source to compare with a small blob of cotton infused with 50 μL n-hexane in the sample bottle as the air source. The 90 males and 90 females were divided into three groups with 30 males and 30 females in each group. Each group of 30 females or males was tested. The experiment was operated according to the ‘gender factor’ experimental method. 

### 2.5. Light–Dark Experiments 

A total of 500 males and 500 females were tested. Ten males or females in each test were placed in two glass Petri dishes (diameter: 15 cm; height: 2.5 cm), and moistened filter paper was placed under the Petri dishes to maintain a level of humidity. The aggregation degree and maximum aggregation at 10 min and 90 min in the light environment (illuminated with a 40 W incandescent lamp) and the dark environment (glass Petri dishes covered with red glass film) number were recorded. This was repeated ten times. After each repetition, the Petri dish was cleaned with 75% ethanol. Additionally, one branch of the Y tube was wrapped with red film, one branch was left untreated, and both sides were illuminated with a 40 W incandescent lamp. Twenty males or females in each test were used to count the criteria for Y tubes in the ‘gender factor’ experiment, and the cleaning tube wall method was used to find the tendency to aggregate in light or dark environments. This was repeated ten times.

### 2.6. Data Analysis

The number of *E. brandti* adults observed in the presence of the host plant and the absence of the host plant was compared using the Mann–Whitney U test and the Kruskal–Wallis test to study the influence of the host on the aggregation tendency. The paired *t*-test was used to compare the influence of the host on the aggregation tendency of males and females. Nonparametric tests (the Mann–Whitney U test and Kruskal–Wallis’s test) were also used to analyze the significance of the *E. brandti* adults’ aggregation in the temperature experiment and light–dark experiment. Moreover, the experimental data obtained through the Y-tube operation were analyzed with the independent sample *t*-test and chi-squared test.

## 3. Results

### 3.1. The Influence of Host Plants on the Aggregation Tendency of E. brandti Adults

A higher proportion of individuals were observed under the host plant shelter. The total number of selected *E. brandti* adults, both male and female, increased with time (Figure 3), and most of them stayed on the host branches. The weevils could be attracted by the host plant, exhibiting an aggregation tendency. In both males and females, the number of weevils that chose a host was significantly higher than the number that chose no host (*p* < 0.001). Moreover, there was no difference between males and females in choosing the host plant in the paired *t*-test analysis (*t* = 0.155, df = 17, *p* > 0.1) (Figure 3). 

### 3.2. Effect of Temperature on Aggregation Behavior and the Tendency of E. brandti Adults

The test results show that aggregation degrees (H = 27.34, *p* < 0.001), max aggregation (H = 59.89, *p* < 0.001), and the number of selected adults (H = 59.25, *p* < 0.001) were significantly different when comparing the independent temperature variable. The aggregation degree was not significantly different between low temperature and high temperature (*X*^2^ = 0.25, *p* = 0.88), but there was a significant difference between low temperature and ordinary temperature (*X*^2^ = 9.14, *p* = 0.01), and between HT and OT (*X*^2^ = 7.19, *p* = 0.03) (Figure 4a). The max aggregation was not significantly different between LT and OT (*X*^2^ = 0.88, *p* = 0.64), but was significant between LT and HT (*X*^2^ = 20.78, *p* < 0.001), and between HT and OT (*X*^2^ = 28.92, *p* < 0.001) (Figure 4b). The number of selected adults was not significantly different between LT and HT (*X*^2^ = 0.02, *p* = 0.99), but was significant between LT and OT (*X*^2^ = 22.72, *p* < 0.001), and between HT and OT (*X*^2^ = 23.46, *p* < 0.001) (Figure 4c). Moreover, we found the aggregation degree was not statistically significantly different (z = −0.930, *p* = 0.352), while the max aggregation (z = −2.018, *p* = 0.044) and the number of selected adults (z = −9.237, *p* < 0.001) were significantly different, using an exact sampling distribution for U (Dineen and Blakesley, 1973). When tested at 10 min and 90 min, aggregation degrees, max aggregation, and the number of selected adults were statistically different.

### 3.3. The Influence of Individual Interaction on the Aggregation Tendency of E. brandti Adults

#### 3.3.1. Genders’ Choice

*Eucryptorrhynchus brandti* adults preferred to choose the odor source over the air (Figure 5). There were significant differences in the response of females (F = −16.005, *p* < 0.001) and males (*t* = 25.732 *p* < 0.001) to air and odor sources (Figure 5a). Meanwhile, there was a significant difference between males and females, and the SRr of males was higher than that of females (Table 1). When male *E. brandti* adults were used as odor sources to attract *E. brandti* adults, there was a significant difference in the response of females to air and odor sources (*t* = 18.760, *p* < 0.001) and also a difference in the response of males to air and odor sources (*t* = 10.404, *p* < 0.001) (Figure 5b), but there was no significant difference between males and females (Table 1).

#### 3.3.2. Responses to Intestinal Crude Extracts

Significant differences were observed in both females (*t* = −61.672, *p* < 0.001) and males (*t* = 39.368, *p* < 0.001) in the response to the crude extract odor sources and to air (Figure 6a). Meanwhile, there was a significant difference between males and females, and the SRr of males was higher than that of females (Table 2). There was a significant difference in the response of females to air and odor sources (*t* = 68.128, *p* < 0.001) and also a difference in the response of males to air and odor sources (*t* = 45.233, *p* < 0.001) when the crude extract of male adults was used as the odor source to attract *E. brandti* adults (Figure 6b), and there were no significant differences between males and females (Table 2).

### 3.4. The Influence of Light Intensity on Aggregation Behavior and Aggregation Tendency of E. brandti Adults

The aggregation degree was not significantly different (z = −1.289, *p* = 0.197) (Figure 7a) and the max aggregation was also not significantly different (z = −1.529, *p* = 0.126) (Figure 7b) when comparing males and females. Both aggregation degree (z = −1.995, *p* = 0.046) and max aggregation (z = −3.564, *p* < 0.001) were statistically different between 10 min and 90 min. Comparing light with dark, the aggregation degree was not statistically significantly different (z = −0.898, *p* = 0.369), while the max aggregation was significantly different (z = −0.274, *p* = 0.784) (Figure 7). There were significant differences in the amount of dark environment and light selected (*t*_male_ = 12.938, *p_male_* < 0.001; *t*_female_ = 11.543, *p_female_* < 0.001), but the number of selected adults was not significantly different between females and males.

## 4. Discussion

We found that host plants could trigger the adults’ aggregation behavior in this study. The number of adults that selected the host plant gradually increased within a certain time range, and most of the adults were found to stay on the host plants in experimental observations. This result showed that the host plant could further trigger the aggregation tendency of *E. brandti* adults [31,32]. In other words, hosts may play a significantly important role in mediating insect behavior through volatile plant signals [33]. A study has shown that the weevil could be attracted by the volatile substance of the tree of heaven [34]. Furthermore, a phenomenon was also observed during the experiment: that males are more susceptible to influences from the host plant, with the number of sheltering individuals increasing with time, but females are only susceptible to the host, and time has little effect. This may be related to the functions of males and females in the groups during the long evolutionary process. In addition to the reproductive role, males may also play the role of searching for food patches. For males, reproduction can be costly in terms of energy expenditure, risk of injury, predation, diseases, and missing the opportunity to look for other mates [35,36]. Males may show behavioral plasticity according to ever-changing physiological demands and environmental conditions and have decision-making circuits.

Environmental heterogeneity, such as temperature and light, can drive the aggregation of the adults, resulting from the environmental preference to seek benefits and avoid harm. The aggregation degree, max aggregation, and the number of adults selected were different in different temperature zones. The weevils had a higher aggregation degree under the low temperature than under ordinary temperature, which indicated that the aggregation intensity of the weevil was slightly smaller, and the aggregation would be dispersed. Under the high temperature, the weevil clusters were larger both in both number and size compared to ordinary temperature, increasing the possibility of forming large clusters. Therefore, the results demonstrated that the weevils tended to cluster more under the low or high temperatures than under the ordinary temperature. *E. brandti* was forced to aggregate at low or high temperatures and select the temperature zone of adaptation when faced with adverse stress of high and low temperatures. However, fire ants *Thermobia domestica* Packard are thermophilic, ranging from 30 to 43 °C, with an optimal temperature of approximately 37 °C, and high temperatures affect many aspects of *T. domestica* such as egg growth and development and microhabitat selection, and clusters are usually found in or near warm areas [22]. In some animals, such as Bonin flying foxes, the ambient temperature was significantly negatively related to the formation of clusters in colonial roosts [21]. However, the max aggregation was not significantly different between low temperature and ordinary temperature, but was significantly different between low temperature and high temperature, and between high temperature and ordinary temperature, which showed that the adaptation regulates the effect of temperature on adults. Although this study did not examine the physiological benefits of clustering at different temperature regions, clustering adaptation may have some metabolic benefits for *E. brandti* adults. Additionally, temperature can cause stress to the activity of *E. brandti* adults; we observed that adult activity was relatively slow at the low temperature, but they became rapidly active at the high temperature. 

There were no significant differences in aggregation degree and maximum aggregation between *E. brandti* females and males under light and dark conditions. *Eucryptorrhynchus brandti* adults gather in both light and dark environments, but the adults clearly preferred the dark environment, whether female or male. Visual cues may play a role in selecting insects to search for suitable sites to perform a range of other behavioral functions [37,38,39]. In a word, *E. brandti* adults can gather in both light and dark environments, which showed that the reason for the aggregation of *E. brandti* adults is not so much related to the difference in light and dark conditions in a narrow space, or the different light conditions is not dominant, but rather it is the signal from conspecific individuals that causes the aggregation. Communication signals between conspecifics include chemical and physical signals, which jointly mediate aggregation behavior [40,41,42]. However, we cannot ignore the fact that it is difficult for us to measure the influence of physical signals such as voice, physical contact, physiological state, and visual signals. Only simple verification of olfactory and chemical signals between individuals is possible. 

In the conspecific interactive experiment, there were significant differences in the response of females and males to air and odor sources. When the female adults or the crude extract of the females were the odor source, the SRr of males was higher than that of females. When the male adults or the crude extract of males were the odor source, SRr of males and females was almost indistinguishable. These results showed that female *E. brandti* could attract male *E. brandti* but not females, and male *E. brandti* could attract both males and females. Similar results were seen in *Cotesia marginiventris* Cresson, in which it was found that the females did not attract or repel each other, but the males preferred the arms in which the females had been released [43]. We identified the mutual attraction of males and females and the chemical attraction of crude extracts of males and females; the experimental results of gender choice were consistent with responses to intestinal crude extracts, thus proving that the olfactory may play an essential role in conspecific individual communication and there also may possibly be a role for chemical signals in communication, with a need to confirm signal components further. Semiochemicals may be a major part of chemical signals. The pheromonal control of aggregation in flour beetles *Tribolium castaneum* Herbst presents one of the best-characterized examples in the Coleoptera [44]. Multiple weevil pheromones have been identified as mediators of individual insect communication, such as *Rhinostomus barbirostris* Fabricius [45] and *Metamasius spinolae* Gyllenhal [46]. The potential use of semi-chemical methods can play an active and effective role in pest management and can be considered a control strategy [47]. 

Currently, pheromone attractants are widely used in pest control. For example, the beard weevil *R. barbirostris* was attracted by the aggregation pheromone released from the male, and a synergistic effect was found when the host volatiles functioned together with the aggregation pheromone [45]. This could provide us with a new way to consider the control of *E.brandti* and develop some more environmentally friendly traps using an attractant mixture of plant volatiles and pheromone. Additionally, pheromone-baited traps would provide an excellent means of identifying biodiversity hotspots, tracking population changes, identifying habitat thresholds for the persistence of target species at the landscape level, and providing feedback to evaluate the effects of conservation management efforts. Monitoring insects based on the exploitation of pheromones or other semiochemicals has the potential to revolutionize the conservation of many insect groups, such as *Osmoderma eremita* Scop. and *Elater ferrugineus* Linnaeus [48]. Meanwhile, the temperature can affect the aggregation behavior and tendency of *E.brandti*, but it is challenging to control *E.brandti* by temperature, and it is not easy to implement. From the point of view of speed and efficiency, the development of attractants is the most convenient means of control.

## 5. Conclusions

In conclusion, interactions between individuals within aggregates and collective decisions are especially relevant as synergistic responses of single individuals can lead to complex and nonintuitive behaviors [49,50,51]. Environmental heterogeneity and homogeneous interaction are the key factors that cause insect aggregation. In this study, we only explored two factors, temperature and illumination; however, other physical signals could not be ignored, and the interaction of physical signals must be the target of our experiments, and the search for chemical signals will also be developed. Studying physical and chemical signals can help us find new control methods and strategies from insects themselves, and lead to effective control and management, which is worthy of further discovery and exploration.

## Figures and Tables

**Figure 1 insects-14-00253-f001:**
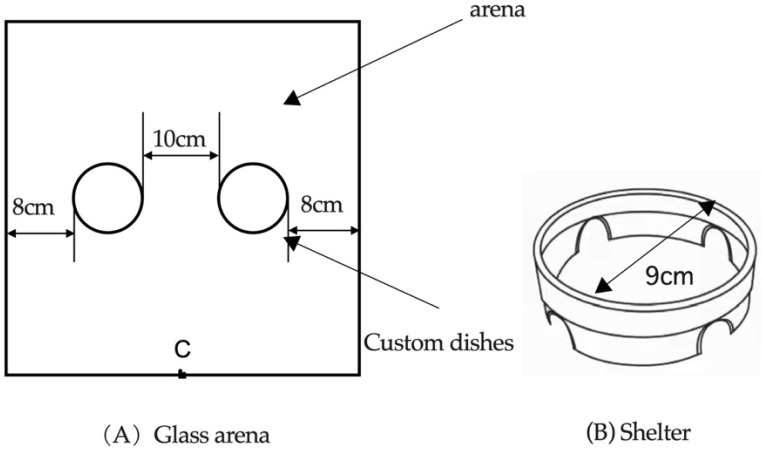
Experimental set-up used in the binary choice experiment. Diagram (**A**) represents the top view of the glass arena and (**B**) represents the shelter.

**Figure 2 insects-14-00253-f002:**
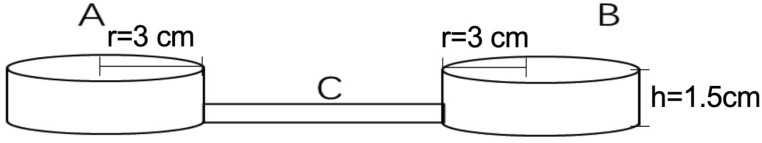
Experimental set-up used in temperature choice experiment. Diagrams (**A**) and (**B**) both represent the glass arena, and (**C**) represents the corridor.

**Figure 3 insects-14-00253-f003:**
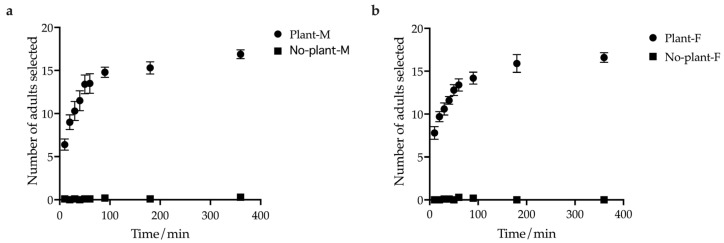
The aggregation tendency of adults with or without the host. (**a**) Variation trend of the males. (**b**) Variation trend of the females. Mean selected number of *E. brandti* found in the presence or absence of host plant shelters for observation times ranging from 10 to 360 min. The results presented here were obtained using a group size of twenty adults, and the variation is expressed by the standard error (n = 10 replicates).

**Figure 4 insects-14-00253-f004:**
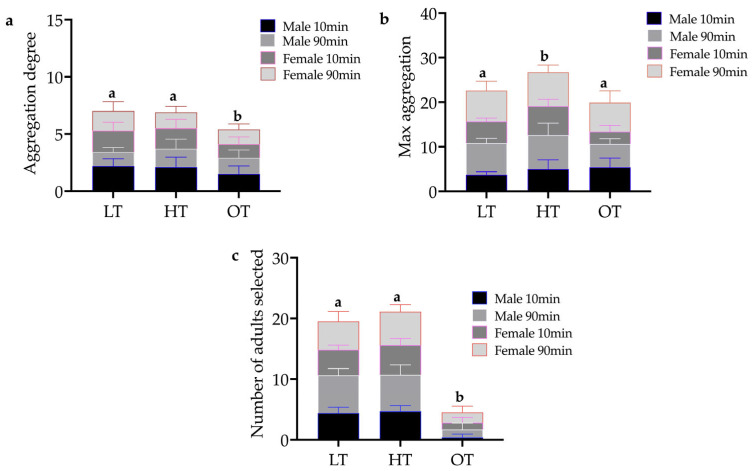
Measures of the aggregation behavior of *Eucryptorrhynchus brandti adults* at different temperatures and two time nodes. (**a**) The number of aggregation degrees. (**b**) The max number of aggregations. (**c**) The *E. brandti* adults choose to enter the ordinary-temperature room at different temperature conditions. (LT, HT, and OT indicate low, high, and ordinary temperatures, respectively). The letters a and b represent the significance label in Figure 4.

**Figure 5 insects-14-00253-f005:**
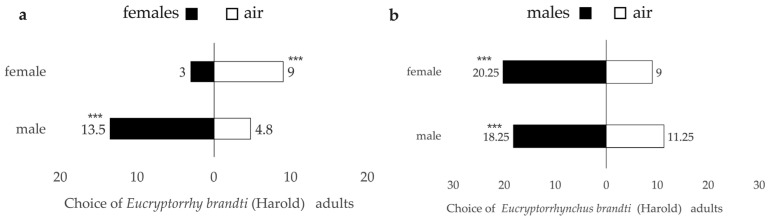
How the *Eucryptorrhynchus Brandt* adults respond to two odorants. (**a**) *Eucryptorrhynchus brandti* adults responding to the female source. (**b**) *Eucryptorrhynchus brandti* adults responding to the male source. *** indicates a chi-squared *p* < 0.001.

**Figure 6 insects-14-00253-f006:**
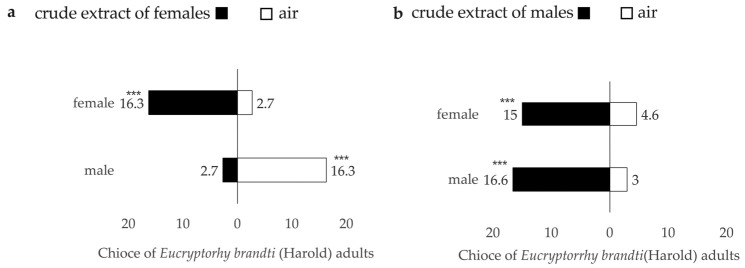
Different responses of the adults to intestinal crude extracts odorants. (**a**) Adults responding to the female intestinal crude extracts. (**b**) Adults responding to the male intestinal crude extracts. *** indicates a chi-squared *p* < 0.001.

**Figure 7 insects-14-00253-f007:**
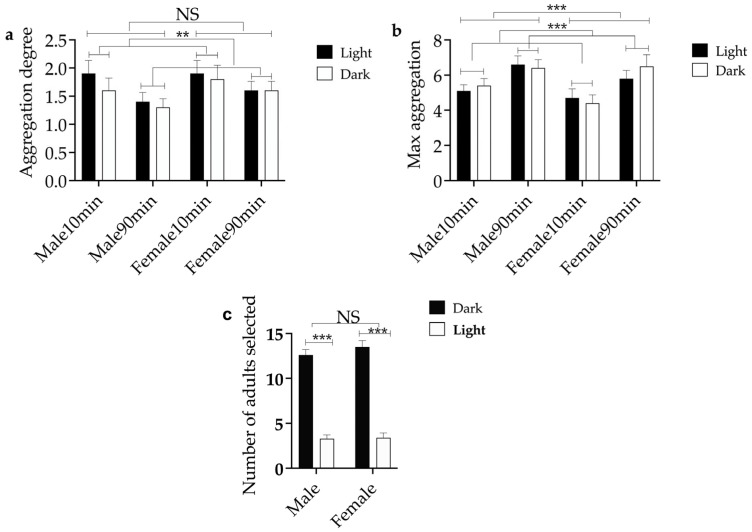
Measures of the aggregation behavior of *E. brandti* with different light conditions and two time nodes. (**a**) The number of aggregation degrees. (**b**) The max number of aggregations. (**c**) The preference of adults for light and dark environments. *** and ** indicates *p* < 0.001, *p* < 0.01, and NS indicates no significance.

**Table 1 insects-14-00253-t001:** Preference in Y-tube containing 20 *Eucryptorrhynchus brandti* adults under different conditions (n = 6).

		Choice of *E. brandti* Adults							
Odor Source Types	Test Insects	Odor Source Tube	Air Tube	Unresponsive Insects	*X^2^*	*p*	^1^ Rr (%)	^1^ SRr (%)	^1^ SC
Female	female	18	54	48	41.496	<0.001 ***	60.00	24.57	−0.5086
	male	81	29	8			91.67	73.74	0.4748
Male	female	79	36	5	0.998	0.339	95.83	68.79	0.3757
	male	75	45	1			99.17	62.24	0.2447

^1^ Rr, SRr, and SC are calculated using Appendix A and indicate response rate, selective response rate, and selective coefficient, respectively. *** indicates a chi-squared *p* < 0.001.

**Table 2 insects-14-00253-t002:** Preference in Y-tube containing 20 *Eucryptorrhynchus brandti* adults under different conditions (n = 3).

		Choice of *E. brandti* Adults							
Odor Source Types	Test Insects	Odor Source Tube	Air Tube	Unresponsive Insects	*X* ^2^	*p*	^1^ Rr (%)	^1^ SRr (%)	^1^ SC
^2^ C. Female	female	8	49	3	58.982	<0.001 ***	95.00	13.97	0.7207
	male	49	8	3			95.00	86.12	0.7224
^2^ C. Male	female	45	14	1	1.352	0.353	98.33	84.82	0.6950
	male	50	9	1			98.33	76.32	0.5263

^1^ Rr, SRr, and SC are calculated using Appendix A and indicate response rate, selective response rate, and selective coefficient, respectively. ^2^ “C. Female or C. Male” means “crude extract from the intestinal tract of females or males”. *** indicates a chi-squared *p* < 0.001.

## Data Availability

Data sharing not applicable.

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
