# Peer review of "The Influencing Factors of Aggregation Behavior of Tree-of-Heaven Trunk Weevil, Eucryptorrhynchus brandti (Harold) (Coleoptera: Curculionidae)"

_insects, 2023, doi:10.3390/insects14030253_

Round 1

Reviewer 1 Report

The manuscript is a description of the results of experimental studies. It is written concisely, research methods are described sufficiently, the results and discussion are also well written. The manuscript has no significant number of complaints.

However, I propose to describe the possibilities of traps that are already being used and developed to combat invasive beetles. Did the authors do their experiments in order to find more ways to control the pest? Then it is necessary to describe the prospects of using such methods.

Also, in conclusion, you need to specify more ways to control the pest.

Author Response

Dear Reviewer,

            We thank you very much for giving us an opportunity to revise our manuscript and pertinent comments on the article. We appreciate the editor and reviewer very much for their positive and constructive comments and suggestions on our manuscript entitled "The influencing factors of aggregation behavior of tree-of-heaven trunk weevil, Eucryptorrhynchus brandti (Harold) (Coleoptera: Curculionidae)" (Manuscript ID: insects-2228803). We have considered the comments very carefully and have revised the paper accordingly. We also modified the grammatical structure and wording of the article with the help of native English speakers. Thank you again for your valuable comments and correct guidance.

Reviewer 1:

The manuscript is a description of the results of experimental studies. It is written concisely, research methods are described sufficiently, the results and discussion are also well written. The manuscript has no significant number of complaints.

However, I propose to describe the possibilities of traps that are already being used and developed to combat invasive beetles. Did the authors do their experiments in order to find more ways to control the pest? Then it is necessary to describe the prospects of using such methods.

Also, in conclusion, you need to specify more ways to control the pest.

Response: Thank you very much for your recognition and valuable comments. We have edited and sorted out this part again, which is placed in the discussion section and modified as follows: “Currently, pheromone attractants are widely used in pest control. For example, the beard weevil Rhinostomus barbirostris was attracted by the aggregation pheromone released from the male, and a synergistic effect was found when the host volatiles functioned together with the aggregation pheromone [1]. This could provide us with a new way to consider the control of E.brandti and develop some more environmentally friendly traps using the mixture attractant of plant volatiles and pheromone. Also, Pheromone-baited traps would provide an excellent means of identifying biodiversity hotspots, tracking population changes, identifying habitat thresholds for the persistence of target species at the landscape level, and providing feedback to evaluate the effects of conservation management efforts. Monitoring insects based on the exploitation of pheromones or other semiochemicals has the potential to revolutionize the conservation of many insect groups, such as Osmoderma eremita Scop. and Elater ferrugineus Linnaeus [2]. Meanwhile, the temperature can affect the aggregation behavior and tendency of E.brandti, but it is difficult to control E.brandti by temperature, and it is not easy to implement. From the point of view of fast and efficiency, the development of attractants is the most convenient way of control”.

  1. Reis, A.C.; Neta, P.L.S.; Jordão, J.P.; Moura, J.I.L.; Vidal, D.M.; Zarbin, P.H.G.; Fávaro, C.F. Aggregation Pheromone of the Bearded Weevil, Rhinostomus barbirostris (Coleoptera: Curculionidae): Identification, Synthesis, Absolute Configuration and Bioactivity. Journal of Chemical Ecology 2018, 44, 463-470, https://doi.org/10.1007/s10886-018-0957-x.
  2. Larsson, M.C.; Svensson, G.P. Monitoring spatiotemporal variation in abundance and dispersal by a pheromone-kairomone system in the threatened saproxylic beetles Osmoderma eremita and Elater ferrugineus. Journal of Insect Conservation 2011, 15, 891-902, https://doi.org/10.1007/s10841-011-9388-5.

Reviewer 2 Report

Review of Sun et al. 

"The influencing factors of aggregation behavior of tree-of-heaven trunk weevil, Eucryptorrhynchus brandti (Harold) (Coleoptera: Curculionidae)".'

I think the authors present sufficient amount of new data with good number of tested insects to be published in Insects.

The major point of revision I found in the manuscript was poor logical flow.

The authors tend to just mention and list previous studies and do not carefully construct logical argument.

The poor logical structure also derives from poor English writing.

I am not a native English speaker myself but I found numerous grammatical errors in the manuscript.

Overall, I would recommend the authors to substantially improve logical structure of the manuscript and resubmit after intensive English editing by professional.

Specific comments are as follows.

Line 12. The second 'Ailanthus altissima' should be abbreviated as 'A. altissima'.

From the second appearance, genus names should be abbreviated.

For example, 'Ailanthus altissima' in line 41 should be abbreviated as 'A. altissima', too.

Exceptionally, genus names should be spelled out at the beginning of sentences.

For example, 'E. brandti' in line 42 should be spelled out as 'Eucryptorrhynchus brandti'.

Carefully re-check scientific names throughout the manuscript.

Line 18-21. This sentence is too long and I did not understand the meaning of the latter half.

Line 44. The sentence ' Ailanthus altissima was introduced into the United State in the 1960s causing serious damage to the local ecosystem and was listed as an invasive species' appears strange and confusing in this paragraph.

This is because you are generally talking about protecting A. altissima from pest weevils but suddenly you start to say A. altissima is a pest plant.

Line 56-70. The two paragraphs are not well organized. For example, the statement in line 56 'Aggregation behavior was observed in some insects and animals' is a bit inconsistent with the later statement in line 59 'Aggregation behavior is widespread in social and non-social insects'.

You say 'some' (= limited number) but later 'widespread' (= many).

You give exampes of insects but not of animals.

You say 'social and non-social' but you don't give any example of social insects. Why dare state 'social and non-social'?

Please carefully choose words and be more logical.

Line 105. Were the test arena replaced for each trial?

I think previous trials can affect later trials if you use the same arena, because of remaining pheromones etc.

Line 113-114. Why did each temperature category had a very wide range (10 degrees)?

Why didn't you just test fixed temperature, for example 15 (cool), 25 (medium), and 35 (hot) degrees?

Besides, the discrimination between ordinary (20-30 degrees) and high temperature (30-40 degrees)  zones are not clear.

30 degrees can be both ordinary and high.

I'm not very sure if it was a sound experiment.

Line 118. The term 'aggregation degree' is confusing and sounds unfamiliar to me. 'Number of clusters' is easier to understand.

Line 130. You should explain earlier that there was 'The filter paper under chamber A and chamber B'.

Line 154. Only 3 replicates is not enough to do statistical ananyses.

I recommend you to increase number of replications, rather than insect individuals in one trial, in your future experiments.

Line 204. Significantly different among what?

Figure 5 and 6. It would become more easy to compare the two tests using live insects (Figure 5) and extracts (Figure 6) if you unify the format of these two figures.

Specifically, in Figure 6a, black bars should come to left side.

Besides, letters in all figures are very small and barely legible.

Line 267-275. I didn't understand this paragraph.

For example, in line 269, I didn't undetsrand 'compared males with females'. Figures 7a and 7b don't seem to compare males and females.

Additionally, in line 270, you state 'statistical differences between 10 mins and 90 mins' but you don't show the data in Figure 7.

Line 285. I don't think this study can justify the argument that 'hosts may play a significantly important role in mediating insect behavior through plant volatile cues'.

The experimental setup didn't have an air flow and the might have been no gradient of odor (volatile cues) within the arena.

The authors didn't control for visual cues.

Line 286. You state that 'A study has shown that the weevil could be attracted by the volatile substance of the tree of heaven'.

Then, what was novel about the present study??

To my mind it sounds like the previous study is sufficient to conclude that hos tree attract weevils and lead to aggregation of the weevils on the host plant.

Line 292. I was not convinced by the argument of the last sentence of this paragraph.

The argument is vague and females may incur the same cost as males.

Line 297. I can't imagine specifically 'resulting in ...'.

Line 299. I don't agree the results in Figure 4 suggest 'E. brandti adults are prone to aggregation in low or high temperatures' compared to ordinary temperature.

Figure 4a and 4b appear to show that E. brandti adults aggregate at all temperature zones and tend to make bigger clusters under ordinary temperature.

Number of clusters is a bit smaller at ordinary temperature than low and high temperatures but this is natural, because the number of total adults is limited and same among all temperature zones.

For example, if you use 50 adults, only 2 clusters of 25 adults are possible, but 5 clusters of 10 adults are possible.

I'm not sure which situation means 'more aggregation'.

Line 328-345. Your results suggest that live females and their extracts repel conspecific females.

I think there have been some similar reports in other insects and females avoid scents of other females to avoid competition among offsprings.

Round 2

Reviewer 2 Report

I would like to appreciate the authors' effort to improve the quality of ms.

Many of my questions have been adequately resolved.

However, I would like to suggest some remaining points to revise before publication.

Overall, I feel the letters in figures are still small and had better be enlarged.

Line 11 and else, TTW. Since the abbreviation TTW is not heavily used in the main text, I would suggest deleting TTW from the whole ms.

Line 12-14. As suggested in the first round review, the sentence of Ailanthus altissima being invasive in non-native range had better be deleted here, too.

Line 82-83. Few but not zero? If so, you should describe what has been done previously, so that the readers can understand what is novel about your study.

Line 96, 109, and perhaps somewhere else. As commented before, genus name at the beginning of sentence should be spelled out. Please carefully re-check the whole ms.

Line 101, 50 cm x 50 cm. This can't be true, because it exceeds the size of experimental arena, which is 44 cm × 44 cm × 3cm.

Line 124-125. This revision is inappropriate and the sentence doesn't make sense. You never explained what chambers A and B are in the previous sentences.

Line 235, aggregation degree. Even though the authors gave a previous literature that used the term "aggregation degree", I don't believe many of the readers can easily imagine the meaning of this term.

Line 309-312, "Furthermore ~ little effect". I assume this sentence refers to some results of this study, not of previous study (ref. 34). However, I cannot find the responding data in any of the figures presented. For example, there is no difference between sexes in Figure 3. Thus, the authors' argument seems wrong.

Line 312-318. Because the previous argument (line 309-3112) is inappropriate, the following argument in line 312-318 should be totally removed.

Line 321, "resulting in". Mistype of "resulting from"?

Line 322, aggregation degree. As commented before, I stick to my opinion that this term had better be avoided.

Line 323-329, "The small ~ high temperatures". I have been confused by this part from the previous version. I think you can write more simply like follows, if my understanding is correct:

Under low temperature, the insects formed a higher number of clusters than under ordinary temperature. The size of cluster was similar between low and ordinary temperatures. Under high temperature, clusters formed by the insects was larger in both number and size compared to ordinary temperature. Therefore, the results demonstrate that the insects tend to cluster more under low or high temperature than under ordinary temperature.

Line 332, "biology temperature". What is biology temperature??
